# Implementation of the Afya conditional cash transfer intervention to retain women in the continuum of care: a mixed-methods process evaluation

Sarah Dickin [1,2] Fedra Vanhuyse [1] Oliver Stirrup,[3] Carla Liera,[1] Andrew Copas [3] Aloyce Odhiambo,[4] Tom Palmer,[3] Hassan Haghparast-Bidgoli [3] Neha Batura [3] Alex Mwaki,[4] Jolene Skordis[3]

¹Stockholm Environment Institute, Stockholm, Sweden
²Department of Women's and Children's Health, Uppsala University, Uppsala, Sweden
³Institute for Global Health, University College London, London, UK
⁴Safe Water and AIDS Project, Kisumu, Kenya

**Correspondence to**
Dr Sarah Dickin;
sarah.dickin@sei.org

## ABSTRACT

**Objectives** We report the results of a mixed-methods process evaluation that aimed to provide insight on the Afya conditional cash transfer (CCT) intervention fidelity and acceptability.

**Intervention, setting and participants** The Afya CCT intervention aimed to retain women in the continuum of maternal healthcare including antenatal care (ANC), delivery at facility and postnatal care (PNC) in Siaya County, Kenya. The cash transfers were delivered using an electronic card reader system at health facilities. It was evaluated in a trial that randomised 48 health facilities to intervention or control, and which found modest increases in attendance for ANC and immunisation appointments, but little effect on delivery at facility and PNC visits.

**Design** A mixed-methods process evaluation was conducted. We used the Afya electronic portal with recorded visits and payments, and reports on use of the electronic card reader system from each healthcare facility to assess fidelity. Focus group interviews with participants (N=5) and one-on-one interviews with participants (N=10) and healthcare staff (N=15) were conducted to assess the acceptability of the intervention. Data analyses were conducted using descriptive statistics and qualitative content analysis, as appropriate.

**Results** Delivery of the Afya CCT intervention was negatively affected by problems with the electronic card reader system and a decrease in adherence to its use over the intervention period by healthcare staff, resulting in low implementation fidelity. Acceptability of cash transfers in the form of mobile transfers was high for participants. Initially, the intervention was acceptable to healthcare staff, especially with respect to improvements in attaining facility targets for ANC visits. However, acceptability was negatively affected by significant delays linked to the card reader system.

**Conclusions** The findings highlight operational challenges in delivering the Afya CCT intervention using the Afya electronic card reader system, and the need for greater technology readiness before further scale-up.

**Trial registration number** NCT03021070.

## STRENGTHS AND LIMITATIONS OF THIS STUDY

⇒ A process evaluation of a cluster randomised controlled trial of a complex maternal health intervention addressing the continuum of care.
⇒ Applies quantitative and qualitative methods to understand how the intervention was implemented in practice.
⇒ Includes views of participants and maternal health nurses who played a key role in delivering the intervention.
⇒ Problems with implementation of the card reader system meant data on participant visits were manually extracted from clinic health records, which were not available for some participants.

## INTRODUCTION

The importance of equitable maternal and newborn health (MNH) is highlighted in the Sustainable Development Goal (SDG) 3: Ensure healthy lives and promote well-being for all at all ages.[1] SDG 3 includes target 3.1 to 'reduce the global maternal mortality ratio to less than 70/100 000 live births' and target 3.2 seeking to 'end preventable deaths of newborns and children under 5 years of age.' Quality maternal healthcare is critical to reach these targets, which includes antenatal care (ANC), delivery at a health facility and postnatal care (PNC).[2] However, a major challenge has been to retain women in the continuum of maternal healthcare to ensure these MNH outcomes can be achieved.

ANC can significantly improve fetal and neonatal outcomes and maternal health in several ways, such as by providing information on good nutrition, prevention and treatment of malaria, management of anaemia and on the benefits of delivery at a facility.[3] Delivery at a high-quality health facility is recommended to address preventable maternal and neonatal deaths.[4–7] In addition, a large

percentage of maternal and neonatal mortality occurs in the postnatal period, which is related to low uptake of PNC services, including danger sign recognition for the mother, and immunisations and promotion of healthy behaviours such as good hygiene and breast feeding for the baby.[8 9]

In Kenya, the maternal mortality ratio and the neonatal mortality ratio remains high with 362/100 000 live births and 39/1000 live births, respectively.[10] Approximately 6 in 10 live births are delivered in a health facility, but only a quarter among women with limited education.[11] Fees for maternal health services were abolished in Kenya in 2013 at all public facilities to improve these outcomes and overcome economic barriers, especially for disadvantaged groups.[12 13] However, despite progress on maternal health outcomes in Kenya, the decline in maternal and child mortality and equity in coverage remains insufficient, with a need to scale up community-level interventions to reach the poor, least educated and rural women, and to ensure they are retained in the continuum of care.[14]

Barriers to the utilisation of maternal services and associated dropout from the continuum of care include poorly equipped health facilities, low quality of care in health facilities and traditional, and religious or cultural practices, among others.[15–17] Distance and travel costs are two other important barriers, as many women are not able to afford the transport costs to seek proper care unless they encounter serious complications during the pregnancy or labour.[18 19] Distance and travel costs can be compounded by poor roads, physical geography and unavailability of ambulances in emergencies.[15] At the same time, increased attendance of visits without increased resources may affect quality of care, due to heavier workloads for healthcare staff, and this has been a challenge in Kenya with the removal of user fees, causing skilled healthcare worker shortages and health worker demotivation.[20]

### Process evaluation aims

This paper describes the results of a mixed-methods process evaluation that aimed to investigate intervention fidelity and acceptability in the Afya conditional cash transfer (CCT) trial. Assessing intervention fidelity is important to accurately interpret intervention outcomes; for instance, an ineffective intervention may be the result of poor fidelity in delivery of the intervention rather than the design.[21] Fidelity is frequently viewed as a multidimensional concept, which includes aspects related to intervention design, provision and receipt.[22] Thus, to assess fidelity collecting as much information on the 'whole picture' is recommended. In particular, measuring receipt of an intervention has been less frequently addressed in health research.[21] In addition to examining how the intervention was delivered compared with its design, this study examined intervention receipt by focusing on assessment of acceptability of the intervention. Acceptability refers to the extent to which people delivering or receiving an intervention consider it to be appropriate.[23] The process evaluation specifically aimed to:

► Investigate factors that affected intervention delivery.
► Assess acceptability of the intervention for participants.
► Assess acceptability of the intervention for healthcare staff.

## MATERIALS AND METHODS

### Study setting

The trial was conducted in 24 intervention and 24 control facilities in Siaya County, Western Kenya. This region has poorer maternal and child mortality than the national average, as infant mortality is 159 deaths per 1000 live births and maternal mortality is 691 deaths per 100 000 live births.[24]

### The Afya CCT intervention

The Afya CCT intervention sought to overcome direct and indirect financial and behavioural barriers to utilisation of maternal healthcare services through the use of cash transfers delivered using the Afya card reader system.[16 25] The aim was to increase use of ANC, delivery at a facility and PNC services compared with the current situation. Increased use of these services, and in particular a continuity of healthcare visits maintained from pregnancy through to 12 months after delivery, is expected to lead to improved MNCH outcomes, including maternal and newborn survival, informed by WHO guidelines for maternal and newborn continuum of care.[3 8]

The Afya CCT intervention was a cluster randomised controlled trial, with equal allocation to intervention and control arms. The units of randomisation were level 2 or 3 health facilities (dispensaries and health centres, respectively).[25] The trial enrolled a total of 2522 women at intervention clinics and 2949 at control clinics. The logic model shown in table 1 shows the intervention inputs, processes and expected changes as the intervention was initially designed. The trial protocol describes the intervention further.[25] Participants were intended to receive cash transfers for each health visit that was attended during pregnancy, childbirth and the postnatal period up to 12 months. During recruitment, clients at the 24 intervention arm health facilities were provided information about the Afya CCT intervention using a short video, before deciding whether to enrol. After enrolment, background information and study arm were registered on an Afya card issued to all participants, and all subsequent visits were intended to be automatically tracked by a nurse who tapped the card using an electronic card reader system, designed for this trial. This included a card reader device with custom-built software, which was enabled for cellular network communication to connect to a remote server, uploading data to the online Afya portal. During periods of limited cellular or electricity connectivity, the card was intended to be used offline with a power bank. Nurses received Ksh400 for each participant enrolled and Ksh100 for each participant at end of intervention period.

By tapping the Afya card at a health visit, cash transfers of Ksh450 were intended to be automatically triggered and made to the participants' mobile money (M-Pesa)

**Table 1** Afya CCT intervention logic model showing inputs, impacts and outcomes

| Intervention inputs | Intervention processes and actions | Changes to direct and indirect financial, cultural and behavioural barriers | Participant immediate impacts | Maternal and child health outcomes |
|---|---|---|---|---|
| Training of nurses on recruitment and delivery of intervention using Afya electronic card reader system | Participant brings Afya card to each appointment | Participants are able to pay for travel to the facility | Participants attend all scheduled maternal health visits, including delivery at facility, as well as immunisation appointments up to 12 months after delivery | Improved MNCH outcomes, including maternal and newborn survival |
| Nurses received Ksh400 for each participant enrolled and Ksh100 at end of intervention period. | Nurse taps card on card reader at each appointment to record visit and trigger M-Pesa or air-time transfer | Visits prioritised by participant and family members, overcoming cultural and financial barriers | | |
| Pregnant women recruited and received Afya card. Cash transfers of Ksh450 for each prenatal, delivery and postnatal visit to health facility in the intervention arm and Ksh50 in the control arm | Monitoring of Afya portal and M-pesa account Reports from the airtime account | | | |
| Afya card reader system available at each control and intervention facility (device hardware and software, sim card for cellular connectivity and online portal) | Electricity at facility or card readers charged and operational during facility clinic hours | | | |
| Phone interviews conducted with participants 1 week after joining, 2 weeks after delivery, 12 months after delivery. | Manual backup system for cash transfers provided by implementation partner | | | |

CCT, conditional cash transfer; MNCH, maternal and child health.

accounts in the intervention arm. This was designed to occur through integration of the card reader system with M-Pesa, a mobile money service in Kenya. Participants who forgot to carry their Afya cards (attached to their clinic books) during the visits or visited non-enrolling facilities that did not have readers installed received manual payouts on verification of the visit by the field implementing partner. In the control arm, the process was similar, with a payment of Ksh50 being made to the mobile phone in the form of airtime.

### Patient and public involvement

Patients or the public were not involved in the design, conduct, reporting or dissemination plans of the research.

### Data collection

To assess the extent to which the Afya CCT intervention was delivered as intended, we focused on the capture of visits using the Afya card reader device and Afya card, and the associated automatic payments triggered. This information was designed to be captured in the Afya portal which was a database of information stored on an external server. Visits were also recorded in participants' clinic books, which is the normal procedure for maternal healthcare at health facilities. These data were later manually collected and inputted into a database at the end of the trial as it was viewed as the most accurate data on visits (see Data supplement 1 of Vanhuyse et al.[26] for more information on this process). However, clinic book data were not available for all participants, particularly some in the control arm, as some participants did not return to the facility for data abstraction at the end of the trial. Data on ANC visits were also recorded in clinic registers, and this data source was available for a large majority of participants for whom clinic book data was not obtained.

To assess the technological challenges linked to the Afya card reader, questionnaires called 'facility reports' were administered over several months from January 2020 to March 2020, using mobile data collection to collect information describing card reader hardware and software problems at every control and intervention facility. Facility staff who completed facility report questionnaires comprised maternal and child health (MCH) nurses (23), nurses-in-charge (NIC) (19) and clinical officers (6). Almost all (47) were involved with the Afya project from its start. Most facility staff members interviewed had 4–5 years' experience (16) and 5 years' experience or more (23), with the remainder with 4 or less years' experience

**Table 2** Characteristics of respondents

| | Nursing role | | Level of facility | | Gender | |
|---|---|---|---|---|---|---|
| | NIC | MCH nurse | Level 2 | Level 3 | Women | Men |
| Nurse respondents (N=15) | 5 | 10 | 7 | 8 | 9 | 6 |
| FGDs with participants (N=5 FGDs) | | | 3 | 2 | 5 | |
| Interviews with participants (N=10) | | | 6 | 4 | 10 | |

FGD, focus group discussion; MCH, maternal and child health; NIC, nurses-in-charge.

(9). A 'spot check' was conducted at 35 facilities in May 2019 to assess adherence to use of the card reader system by healthcare staff.

Acceptability of the intervention for participants and healthcare staff was assessed using qualitative methods through one-to-one interviews and focus group discussions (FGDs). All interview and FGDs topic guides were developed by the research team and pretested in the field. We conducted semistructured one-to-one interviews with 15 nurses, including NIC and MCH nurses (table 2).

Interview guides included questions related to the acceptability of the Afya CCT intervention components, including use of the Afya card reader system, cash transfer method and how delivering the Afya CCT intervention impacted workload and job performance. Interviews with nurses were conducted in English, were digitally recorded with participants' consent and transcribed verbatim. We conducted 10 semistructured one-on-one interviews with intervention participants and five FGDs, with approximately 8–10 participants in each FGD, sampled to cover different facility types (table 2). Interviews and FGDs with participants focused on the acceptability of receiving cash transfers for visits, including the mobile money format, perceptions of maternal health services received, and attitudes of the participants and their families towards the effectiveness of the cash transfer, including experiences of the delays encountered. All interviews and FGDs were conducted between February 2019 and April 2019. These interviews and FGDs with participants were conducted in Luo, the local language. All interviews were digitally recorded with consent of participants, translated and transcribed verbatim in English.

### Data analysis

Data processing and descriptive analyses were conducted using Stata (V.15.1). Graphical summaries were generated using the ggplot2 package (V.3.3.2) for R (V.4.0.2). Quantitative data were analysed descriptively, using summary statistics, and were compared with data recorded in clinic books. Qualitative data were analysed thematically using framework analysis that was conducted in Microsoft Excel.[27] First, transcripts were read several times, and then transcripts were coded guided by factors affecting acceptability of health interventions developed by Sekhon et al,[23] including perceived burden, perceived effectiveness, ethicality, intervention coherence and affective attitude. Framework analysis involved developing a series of

matrices in Microsoft Excel to organise data by theme and subtheme.

### RESULTS

Results are presented in two sections comprising intervention delivery and acceptability of the intervention. A description of participants is included in the associated impact paper.[26]

### Intervention delivery

#### Visits and payments registered in the Afya portal

Overall, a low proportion of visits were captured as intended. In the Afya portal, 6440 of the 25 085 (26%) total registered payments were recorded automatically, indicating that the visit was recorded using the card reader device and Afya card, and that a transfer occurred as intended. The remaining 74% of payments required involvement of the field implementation partner to manually record a visit and trigger a transfer. For this to happen, a participant or nurse needed to place a phone call to the field implementation partner and report a visit.

There was a substantial drop-off in the registration of automatic payments in the Afya portal over time (figure 1). Most automatically recorded visits and transfers took place in 2017 and 2018, and in the later part of 2018 automatically recorded visits dropped off. Compared with 27% of visits in 2018, only 3% of visits were captured in the Afya portal in 2019 for the intervention arm, indicating a large decrease in adherence to the intervention towards the end of 2018 (table 3). This also corresponded to the period after enrolment was completed, and healthcare staff no longer received incentives until the end of the trial.

Automatic payments are also an indication of timeliness. Transfers that required manual triggering (74%) were delayed, often by months. At the end of the trial, during the clinic book data abstraction, any missing payments were transferred to participants. Incentives for healthcare workers were completed manually using M-Pesa (table 1).

#### Facility reports on use of the Afya card reader system

All facilities reported problems with the Afya card reader system, including receiving error messages on the card reader device screen when the card was tapped on the card reader (at all facilities), the system not responding

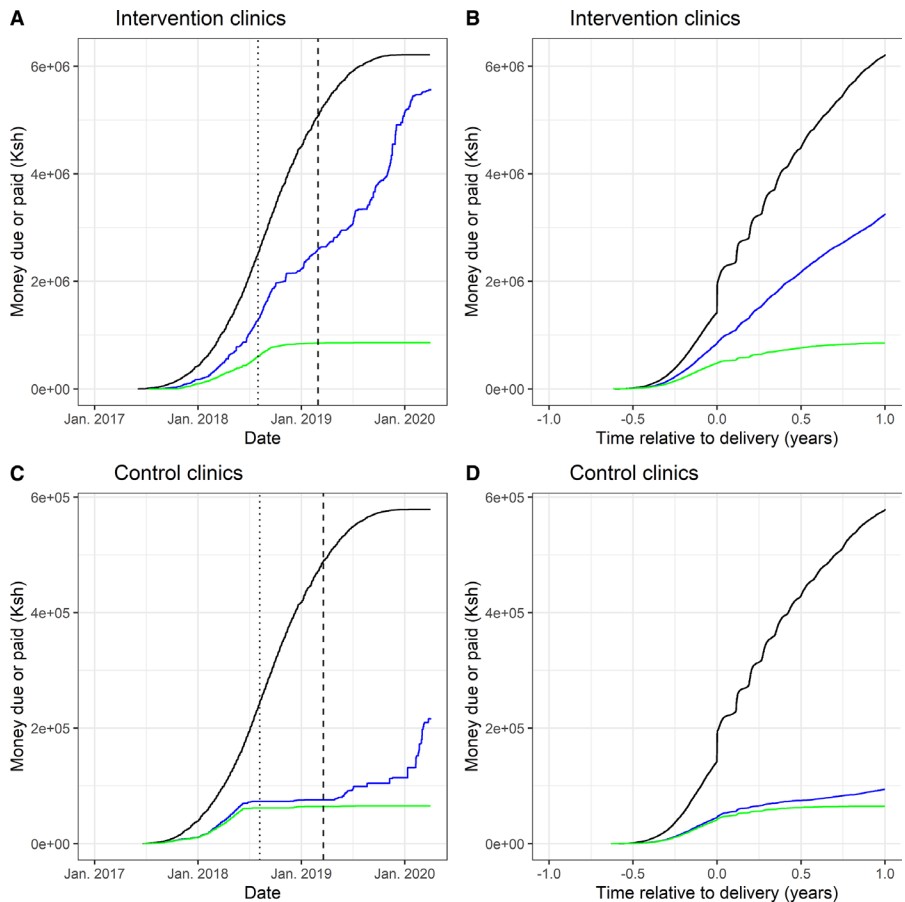

**Figure 1** Cumulative planned and completed payments made to participants among the subset with clinic book data available. The black curve shows money due while the blue and green lines indicate total money paid and money paid through the intended automated M-Pesa system linked to the Afya portal, respectively. The dotted line shows data of last enrolment of a participant, and the dashed line indicates latest recorded date of delivery in the group. (A) Payments made in intervention facilities by date; (B) Payments made in intervention facilities by time relative to delivery (C) Payments made in control facilities by date; (D) Payments made in control facilities by time relative to delivery.

when a card was tapped (34 facilities) and incorrect names associated with the cards of the participants (at 10 facilities). Problems were also reported (29 facilities) during card replacement as most participants could not recall their card numbers. Afya portal data indicated that 539 participants had 'spoilt cards' which were replaced, which means that approximately 10% of cards were spoilt among the 5488 participants.

The most common challenge reported with the device itself was that the screen froze during use (34 facilities), would not charge (32 facilities) or would not work due to poor network connection (30 facilities) (table 4). At 23 facilities, the card reader device was picked up for repair, which required it to be sent to Nairobi. Reported challenges with power banks, which were designed to provide a back-up power supply, included power draining too fast (28 facilities) and power banks being lost (1 facility).

### Spot check of use of the Afya card reader system

During spot checks in 2019, it was found that only four facilities had the Afya card readers charged and in use. On testing, all card reader devices were functional except for one which could not be charged. This card

reader device was removed and replaced the following day. Healthcare staff reported several reasons for not adhering to the intervention including: believing that the intervention had ended; not tapping the cards to avoid conflicts with participants over delayed transfers due to the faulty system, including sometimes telling participants the trial had ended; new staff at facilities who did not want to participate due to challenges with delayed transfers; no training for new staff; and the card reader being locked by the main staff member actively involved in the intervention to avoid theft but limiting the use by other healthcare staff.

### Context

During the Afya CCT intervention enrolment period in 2017, there was a prolonged nurses' strike of 150 days that occurred in Siaya county and elsewhere in Kenya, followed by shorter strikes later in the trial.[28] This resulted in paused enrolment of participants in the intervention. In the facility reports, nurses reported strikes occurring during the intervention at 45 facilities. During the intervention, there was also a change of regulations in Kenya related to M-Pesa cash transfers, which required

**Table 3** Visits made compared with visits recorded by tapping the Afya card*

| No of visits | Intervention arm | | Control arm | |
|---|---|---|---|---|
| | Clinic book | Portal (% of clinic book) | Clinic book | Portal (% of clinic book) |
| **Year** | | | | |
| 2017 | 1508 | 354 (23) | 1104 | 230 (21) |
| 2018 | 10 804 | 2956 (27) | 8382 | 1549 (18) |
| 2019 | 5382 | 155 (3) | 4295 | 23 (1) |
| 2020 | 59 | 0 (0) | 41 | 0 (0) |
| **Facility level** | | | | |
| 2 | 12 703 | 2456 (19) | 9543 | 1246 (13) |
| 3 | 5035 | 1008 (20) | 4263 | 549 (13) |
| Total† | 17 754 | 3465 (20) | 13 835 | 1802 (13) |

*Total values are slightly larger than sum across years and facility levels due to missing data.
†These data are a subset of the overall dataset of portal visits, as it corresponds only to women who brought their clinic book, allowing complete visit data for the comparison. For some participants, especially in the control arm, it was not possible to obtain the clinic book data for comparison. Repeat visits on the same day are not included.

reauthorisation of the field implementation partner, creating a short-term stop to transfers. In addition to these events, interviews with healthcare staff indicated that the study location is characterised by existing labour shortages and limited resources, with a high work burden for nurses.

### Acceptability
#### Acceptability by participants
Acceptability by participants was mainly impacted by the ways that the intervention delivery deviated from the planned activities. According to facility reports, all surveyed facilities received complaints from participants related to delays with payment of cash transfers.

#### Perceived effectiveness
Participants were enthusiastic to join the trial, and when participants received the cash transfer, they were happy with the intervention and reported it allowed them to pay for transport. This was especially important in cases where they were referred to another district, for example, for a scan. Increased PNC visits were reported, as one participant described: 'Before we used to go when there was an injection that was to be given to the child but if it was only the weight that was to be taken then I did not see the importance of going.'

However, most participants described a situation where their cash transfers for attending appointments were not received automatically at the facility visits. Participants then asked nurses for help or called the number on the Afya card, which resulted in receiving a payment, as it was then processed manually by the field implementation partner. However, due to the large volume of manual payments, these were often delayed. One participant described waiting at the clinic a long time for money to arrive to pay transport fare but did not receive it. Participants were satisfied with receiving the payment in the M-Pesa format as they found it more convenient than having to come at a particular time to pick-up cash. They also believed that money in the form of cash would not be distributed correctly and knew that with M-Pesa they were getting the full amount.

#### Ethicality
Participants reported that their family members were supportive of the programme. Some felt that no one should be paid to attend appointments, and that it could create problems with future attendance. Rarely, the cash transfers introduced ethical problems for participants who were asked to buy something for healthcare staff, such as a drink: 'They will tell you to give them something…that they are the ones who have made you get that money and that you should buy for them a soda or give them something small.' Some participants reported that

**Table 4** Overview of reported problems with Afya card reader device per intervention arm by facility

| Reported problems with card readers | Control arm facility | Intervention arm facility | Total facilities |
|---|---|---|---|
| Sometimes the screen would freeze | 20 | 14 | 34 |
| The card reader device would not charge | 18 | 14 | 32 |
| Poor network connectivity at the facility | 17 | 13 | 30 |
| System failure message | 18 | 11 | 29 |
| Card reader rapidly lost power | 17 | 11 | 28 |
| Waiting for the card reader to be picked up and repaired took a long time | 13 | 10 | 23 |
| The card reader displayed the introductory video but the user could not proceed with enrolment | 12 | 9 | 21 |
| The card reader switched off automatically and would not power on | 14 | 6 | 20 |
| Card reader charger got lost | 0 | 1 | 1 |

the transfers changed intra-household dynamics, such as a spouse expecting a share of the money to provide transport or changing household distributions of income.

## Acceptability by nurses
### Affective attitude and intervention coherence
The aim of the Afya CCT was clear to nurses, who understood that the intervention aimed to increase use of maternal health services. Nurses reported many challenges with implementation of the intervention, including inconsistent and variable success with automated payments. This impacted acceptability because nurses were viewed as the 'face' of the Afya CCT intervention to clients and were worried about damaging their community relationships. Nurses reported limited opportunity to give feedback in the intervention design, to exchange experiences with other facilities running the intervention or to receive performance feedback on their involvement.

### Perceived effectiveness
Nurses perceived improvement in their facilities' service delivery figures, especially ANC and immunisations, because of the Afya CCT intervention. Increased attendance for ANC resulted in hitting targets set by the county government as one nurse described: 'My targets are good and at least we are shining.' These targets included percentage of clients attending their fourth ANC visit, which was rarely attained prior to the Afya intervention. As this plays a role in the nurses' performance reviews, with potential for greater funding and staff, and better relationships with county government, many nurses recommended scaling-up the programme. Nurses reported that clients started coming for ANC visits earlier than usual with the Afya CCT intervention, as well as fewer missed immunisation visits: 'There was an increase in the number of mothers attending the fourth ANC, and the defaulters in terms of immunisation was reduced.' For some nurses perceived effectiveness was negatively impacted by card reader issues and they were less likely to use the Afya card reader. One nurse described low interest in tapping the cards: 'But now they are not getting the funds, yes…They are saying 'now this thing…' some of them even have the card but they do not want to produce the card, because it does not make any difference.'

### Burden and opportunity cost
For the most part, the Afya CCT intervention increased nurses' workload. An increase in attendance and the failures of the Afya card reader system resulted in a greater burden of work for nurses. One nurse described the workload increase linked to Afya in the context of scaling-up the intervention: 'I would tell them to be very ready for this programme, because it requires a lot from them.'

An increase in workload was related to a greater number of clients attending the facilities for maternal health services, as well as workload related to carrying out administrative tasks in the Afya CCT intervention, particularly those related to Afya card reader failures. Nurses described fewer clients missing their ANC appointments, increasing workload. The greatest administrative burden was during the enrolment period, including showing possible participants a video, enrolment paperwork, such as entering information into the card reader and on paper, and time needed to learn how the intervention worked. Nurses described an increase in queues, exacerbating long wait times. The monetary incentive (Ksh400) for each enrolled participant was reported as too small by some nurses compared with the additional work that was required, and that incentives should have continued throughout.

With card reader failures, a large burden of time was needed to interact with the implementing field partner to replace card readers, register new cards and help participants to resolve their lack of payments. Nurses also received many complaints from participants. Some nurses reported retaining the participants' cards and registering the visits when the network connection was working, which could be outside their working hours. A nurse described the reality of these delays: '…payment of money is taking too long and these clients were told that this is money they would be receiving every month. You find that a client has come for all the ANC visits, all the immunisation services and now the baby is 7 months with no single cent.' In some ways the cash transfer reduced workload, as participants would be more likely to come at the right time and day, reducing overloading at certain times, or were more patient in waiting in long queues.

### Ethicality
Many nurses had objections to cash transfers being used as an incentive and suggested the incentive could instead be material baby related items, for example, blankets. They believed this would directly benefit the child, whereas the cash transfer could be used for other things. Some nurses had ethical concerns related to using cash to motivate participants to attend maternal health visits, especially as visits were free of charge. For these nurses, they viewed it as the responsibility of the client to seek healthcare services. This was also seen as setting a precedent that could create future problems: 'They should take it that it's their role to attend the services. If they come and get the services free of charge, why should they be paid to get the services?…If the health worker is there and the health worker is attending to them, why should they be paid to get the services? I don't think it's something that we can sustain.' When problems with the card readers occurred many nurses felt responsible for letting participants down and were uncomfortable dealing with participant complaints: 'I will understand that the system failed but what about the client? The client is seeing it as if it is me who has eaten that money, which is not the case.'

Some nurses believed that occasionally participants wanted to secure a spot in the Afya CCT intervention, trying to enrol even before their pregnancy could be

confirmed, indicating possible unintended consequences: 'Someone at 3 weeks' time, the PDT (pregnancy test) is positive; the client is here and wants to start ANC. You are not even sure of the pregnancy and the client is insisting.' In addition, nurses reported that a few participants had travelled to an intervention arm facility, even if they were inside the catchment area of another facility or switched from a control to an intervention arm facility.

## DISCUSSION

Several studies have examined the use of CCTs in the context of improving the use of maternal healthcare,[7 29–34] but there has been less focus on the role of technological innovations for this purpose. The Afya CCT intervention was a complex intervention with multiple interacting components that included delivery of CCTs using the novel Afya card reader system that triggered automated mobile money payments. Both nurses and clients who participated in the trial found the intended cash transfer system using the Afya card linked to M-Pesa to be acceptable when it worked as intended. Results of the impact study indicated that the intervention arm was slightly more likely to be retained in ANC and to attend immunisation appointments, but had no effect on delivery at the facility or on attendance of at least one PNC visit.[26] In line with this, nurses perceived the intervention to be effective at reaching targeted numbers of ANC visits, which were a challenge to achieve prior to the trial, as well as improving immunisation visits. While nurses perceived that clients came earlier for the first ANC visit, there was no difference between the intervention and controls arms in mean gestational age at first ANC visit.[26] However, other aspects of acceptability were lower from the perspective of nurses, including an extra burden of work created by the intervention, and ethical concerns related to cash transfers for maternal health visits. Some nurses viewed attendance at visits as part of the responsibility of the clients, and similar findings related to personal responsibility have been reported in a financial incentive intervention in Johannesburg, South Africa, to improve retention in HIV care among pregnant women.[35]

Overall, the Afya CCT intervention was characterised by low fidelity in terms of automated payments and registration of visits using the Afya card reader system, and low adherence to the use of the card reader system by healthcare staff. Only 26% of visits were automatically registered with a successful transfer triggered, meaning other transfers were delayed due to the need for manual processing. This had no one simple explanation but major issues were likely related to failures of the card reader system with problems reported at each facility, to low adherence to use of the card reader system, particularly after the enrolment period when incentives for nurses ended, and to contextual challenges in the study site.

A key challenge with the delivery of the intervention was that nurses reported being the 'face' of the Afya CCT intervention to participants when issues occurred,

decreasing adherence to use of the card reader system over time to avoid additional work related to manual transfers and damaging client relationships. Other interventions aiming to increase use of MCH services or other health services have highlighted the attention required to the implications for exacerbating existing staff challenges.[36] A number of non-financial strategies have been found to motivate health staff to improve delivery of interventions, such as recognition of their contributions and ensuring adequate resources and appropriate infrastructure.[37 38] Other CCT studies have used financial incentives for healthcare workers as well as clients to improve outcomes, for example, a CCT programme in India that significantly increased the use of maternal healthcare services found that larger incentives for health workers were associated with higher utilisation rates compared with larger incentives for mothers.[39] If nurses had received cash transfers for each client visit, this may have led to greater use of the card reader system in the Afya CCT intervention.

The findings indicate several considerations and operational requirements for delivering the intervention at scale. A lack of sustainability has been a common problem with implementation of IT-based health solutions in low-resource settings.[40] In the case of the Afya CCT a much more robust technological system would be needed for visit data collection and distribution of cash transfers that could operate without breakdowns and ensure transfers only went to intended recipients. Adequate training and ongoing monitoring and feedback for healthcare staff is also important, such as effective back-up systems that healthcare staff have been trained on. Prior to further scale up a greater technology readiness level (TRL) is needed. Although not widely used in low-income settings, TRL has been used to assess new health technologies such as telemedicine in Uganda.[41 42] There are also opportunities to learn from the strengths and weaknesses of other electronic cash transfer programmes and card reader systems, such as Aadhaar cards in India which have been implemented on a larger scale. Aadhaar records biometric data including ten fingerprints, two iris scans and a facial photograph to ensure benefits reach intended recipients.[43] Aadhaar has demonstrated the potential for collecting data to greatly improve the planning and delivery of public health interventions, and also highlighted the need to ensure safe use of patient data and the ways such systems can have severe negative consequences when they malfunction.[44 45] Alongside technological improvements, an adequate framework for ensuring data privacy and public trust is needed to mitigate associated risks prior to further scale up.[44]

This study includes several limitations. In the study design, problems with the card reader system and use of manual back-up for payments were envisioned as a rare occurrence. In light of the more frequent challenges faced in implementation, facility reports and spot checks were added to the study design that were not in the original protocol.[25] More regular reports of how the card

reader system was being used throughout the entire trial period would have provided more detailed insight on the types of problems with card reader use as well as their timing and frequency. We also did not collect broader contextual information on individual health facilities, such as level of remoteness or frequency of electricity or network outages, that could have provided insight on which facilities would need more field support to implement the intervention.

## CONCLUSIONS

Design and delivery of interventions become more challenging as they move from simple towards complex interventions with multiple components and interactions. In the Afya CCT intervention to retain women in the continuum of maternal healthcare, healthcare staff and clients found the delivery of cash transfers using mobile money payments to be acceptable when it worked as intended, such as increasing ANC visits which had been difficult to achieve. However, the Afya CCT intervention was complex, and characterised by low fidelity in terms of automated payments and registration of visits using the Afya card reader system. This contributed to low adherence to the use of the card reader system over the entire trial period. Better technology validation and demonstration is needed prior to further scale up of the Afya CCT intervention or similar interventions relying on electronic card reader systems and automated payments.

**Acknowledgements** The authors thank the following people for their contributions to the development of the intervention and the design and execution of this study: Katherine Harrison, Caroline Ochieng, Alie Eleveld, Geordan Shannon, Stacey Noel, Matthew Fielding and Sangoro Onyango. The authors are grateful to all participating nurses who delivered the Afya intervention.

**Contributors** SD led the process evaluation with support from FV, JS-W, NB, HH-B, TP and AC who contributed to the research design and methodology. CL contributed to analysis of qualitative data. OS, TP and AC contributed to data analysis of quantitative data. AO was the trial coordinator, leading field implementation under supervision of AM. All authors contributed to the interpretation of the research findings and the writing of the paper. All authors read and approved the final manuscript, SD is guarantor.

**Funding** This work was supported by the Bill & Melinda Gates Foundation (grant number: OPP 1142564) and the Swedish International Development Cooperation Agency who provided cofinancing to SEI (grant number: not applicable).

**Disclaimer** The funders were not involved in the design, implementation, data collection, analysis, writing and decision to submit the paper for publication.

**Competing interests** The authors declare no competing interests, aside from AC who is associate editor of Sexually Transmitted Infections.

**Patient and public involvement** Patients and/or the public were not involved in the design, or conduct, or reporting, or dissemination plans of this research.

**Patient consent for publication** Not applicable.

**Ethics approval** Ethics approval was given by Maseno University Ethics Committee, Kenya (Reference number SU/DRP/MUERC/00294/16). Participants gave informed consent to participate in the study before taking part.

**Provenance and peer review** Not commissioned; externally peer reviewed.

**Data availability statement** Data are available on reasonable request. Deidentified data may be made available on request to researchers who provide a scientifically and methodologically sound research proposal and obtain ethical approval for their planned analysis. Proposals should be submitted to the corresponding author. Data that can be shared includes number of visits made, type of visit, arm and date of the visit. Data dictionaries can be made available, as well as the study protocol and the statistical analysis plan. Data are available now.

**ORCID iDs**
Sarah Dickin http://orcid.org/0000-0003-0437-3755
Fedra Vanhuyse http://orcid.org/0000-0002-9283-1914
Andrew Copas http://orcid.org/0000-0001-8968-5963
Hassan Haghparast-Bidgoli http://orcid.org/0000-0001-6365-2944
Neha Batura http://orcid.org/0000-0002-8175-8125

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
