## [Reviewer comments · BMJ Open]

ARTICLE DETAILS

TITLE (PROVISIONAL)	Implementation of the Afya conditional cash transfer intervention to retain women in the continuum of care: a mixed-methods process evaluation
AUTHORS	Dickin, Sarah; Vanhuysse, Fedra; Stirrup, Oliver; Liera, Carla; Copas, Andrew; Odhiambo, Aloyce; Palmer, Tom; Haghparast-Bidgoli, Hassan; Batura, Neha; Mwaki, Alex; Skordis-Worrall, Jolene

VERSION 1 – REVIEW

REVIEWER	Wood, Francine Tulane University School of Public Health and Tropical Medicine
REVIEW RETURNED	31-Jan-2022

GENERAL COMMENTS	r outages or internet connectivity issues. - Were the facility reports conducted only in the intervention areas? If there were in both study areas, how many were conducted in each area? Was it an equal distribution and how many from each facility? Including this information may help make Table 3 easier to understand. Discussion - I would challenge the authors to reevaluate some of their considerations and operational suggestions for delivering interventions. This section addresses some of the challenges presented but not many. It would be beneficial to identify how the different issues identified from the study be addressed. For instance, how can the issue with the delays with the payments be addressed and where does the issue originate from? Do the Aadhar cards described fix this issue or just ensure that the intended recipient gets the funds? Does it address the issue mentioned in the article about the other expecting a portion of the refunds? What can be done to address these challenges?- There was no discussion of the study's limitations. Other - There were several spelling or grammatical errors that the authors should review.- In the discussion section, the citation format changed. Authors should fix this.
---

REVIEWER	Clouse, Kate Vanderbilt University School of Nursing
REVIEW RETURNED	09-Feb-2022

GENERAL COMMENTS	The authors present a clearly written manuscript describing the results of a mixed-methods process evaluation to investigate fidelity to and acceptability of a conditional cash transfer (CCT) intervention in Kenya. Overall, the authors found disappointing results with fidelity and serious implementation concerns that limited the success of the study. It is important for negative results like these to be shared so that other researchers may learn from these experiences. The paper has already published earlier manuscripts and in an attempt to avoid duplication, this manuscript leaves out some important details in the Methods relevant to the current study. On page 7 in Study Setting, the authors should note that this was a cluster-randomized trial and give the number of facilities involved. Table 1 and other parts of the manuscript allude to nurses receiving CCT, but this is not properly explained in the description of the intervention. Describe this more. Table 3 should note that these numbers are by facility. The Results section spends ample space discussing nurses' views, but the Discussion fails to fully discuss these results. It is summarized that the intervention was acceptable to nurses, but the Results suggest that this was more nuanced, particularly in increased workload and the section "Ethicality." The findings in the latter section are resonant of those in another CCT among pregnant women in South Africa. See: Acceptability and feasibility of a financial incentive intervention to improve retention in HIV care among pregnant women in Johannesburg, South Africa. AIDS Care 2018 Apr;30(4):453-460. Minor edits: - Page 16, line 8: add a comma after "testing" - Page 19, line 57: "increase" not "increased" - Page 20, line 8: add "to" between "needed interact"
--

VERSION 1 – AUTHOR RESPONSE

Reviewer 1

Abstract - As written, the sentence starting on line 43 is unclear and the authors should consider rephrasing it.	This sentence has been rephrased: "Delivery of the Afya CCT intervention was negatively affected by problems with the electronic card reader system and a decrease in adherence to its use over the intervention period by healthcare staff, resulting in low implementation fidelity."
Introduction - Authors may want to consider rewriting the introduction to make the connection between the health problem more evident. It's more apparent in the methods section, but this needs to be in the introduction. How was the Afya conditional cash transfer related to the health problem	The intervention is focused on retaining women in the continuum of maternal healthcare including antenatal care (ANC), delivery at facility and postnatal care (PNC), with the overall goal to improve maternal and newborn health outcomes. We have added a few more references to the continuum of care concept in the introduction to

and what was the purpose? I think this vital connection is missing and should be introduced briefly before the section on the process evaluation aims.	clarify the link between the intervention purpose and this health challenge (assuming this is what the reviewer means by health problem).
Page 7 on line 55, consider omitting the word “briefly”	Briefly was removed
Page 9 on line 5, review punctuation for the sentence starting with “After enrolment...” Should there be a comma before “background information...?”	Comma inserted
It will be informative if the background characteristics of the interview and FGD participants are included in the paper. The impact paper referenced describes participants in the entire study.	We have added information on the facility level, gender and role (nurses) for interviews and focus group discussions with nurses and participants. In the description of the intervention we have also differentiated between level 2 (dispensary) and level 3 (dispensary).
After several reads, I gathered that interviews were conducted with nurses, and these were in English. The other interviews and FGDs with intervention participants were in Luo. If this is correct, consider rephrasing or combining the following sentences “We conducted semi-structured one-to-one interviews with 15 nurses” and “Interviews were conducted in English, were digitally recorded participants consent and transcribed verbatim.” Secondly, were the 10 semi-structured interviews conducted with intervention participants one-on-one? The authors should clearly state this since it was done for the interview with nurses.	Yes, this is correct. We have clarified this distinction in the text with regard to the interview language for the nurses (English) and the trial participants (Luo). We have also added ‘one-on-one’ to description of interviews with participants.
State the names and versions of the statistical programs in which the quantitative and qualitative analyses were conducted	We have added details about software used for data analysis in the text. Data processing and descriptive analyses were conducted using Stata (version 15.1). Graphical summaries were generated using the ggplot2 package (version 3.3.2) for R (version 4.0.2) and framework synthesis was conducted in Microsoft Excel.
It will be insightful to answer the following question. What were the different facility types and did any of the results (e.g., intervention delivery) vary depending on the type of facility? I can imagine the facility's location may affect if the visits were recorded	We have dis-aggregated results by the level of healthcare facility, and added information about what the facility level refers to. We do not have data to dis-aggregate the results in other ways such as power outages.

as intended. For instance, if there were frequent power outages or internet connectivity issues.	
Were the facility reports conducted only in the intervention areas? If there were in both study areas, how many were conducted in each area? Was it an equal distribution and how many from each facility? Including this information may help make Table 3 easier to understand.	The facility reports were collected in all control and intervention facilities, we have added this clarification to the section on data collection. Only one report was conducted in each control and intervention facility, so 48 facility reports were collected from nurses to report problems encountered during the trial. There was also a separate spot check to observe use of the card reader systems. Table 3 shows the break-down of problems reported for the two types of facilities.
I would challenge the authors to reevaluate some of their considerations and operational suggestions for delivering interventions. This section addresses some of the challenges presented but not many. It would be beneficial to identify how the different issues identified from the study be addressed. For instance, how can the issue with the delays with the payments be addressed and where does the issue originate from? Do the Aadhar cards described fix this issue or just ensure that the intended recipient gets the funds? Does it address the issue mentioned in the article about the other expecting a portion of the refunds? What can be done to address these challenges?	There is limited space in the discussion to discuss all the possible solutions to fix the card reader system due to word count, which is why we have generally focused on better testing and technology readiness level overall, which would have reduced many of the interconnected challenges. The delays were all due to manual payments needing to be made and a resulting backlog, as a result of problems with the card reader system. We have clarified this in the discussion: “Only 26% of visits were automatically registered with a successful transfer triggered, meaning other transfers were delayed due to the need for manual processing. This had no one simple explanation and likely related failures of the card reader system with problems reported at each facility, to low adherence of use of the card reader system, particularly after the enrolment period when incentives for nurses ended, and to contextual challenges in the study site.” We have also added further reference to recent CCT interventions that have used incentives for health care workers to improve outcomes, as this could address some of the issues with nurses being motivated to use the technology: “Other conditional cash transfer studies have used financial incentives for health care workers as well as clients to improve outcomes, for example, a CCT program in India that significantly increased the use of maternal health care services found that larger incentives for health workers were associated with higher utilization rates compared with larger incentives for mothers [39]. If nurses had received cash transfers for each client visit,

	this may have led to greater use of the card reader system in the Afya CCT intervention.”
There was no discussion of the study’s limitations.	A paragraph has been added to the end of the discussion providing an overview of some key limitations of the study.
There were several spelling or grammatical errors that the authors should review.	We have proof-read the revised manuscript for spelling and grammatical errors.
In the discussion section, the citation format changed. Authors should fix this.	The citation format in the discussion has been fixed.

Reviewer 2

The paper has already published earlier manuscripts and in an attempt to avoid duplication, this manuscript leaves out some important details in the Methods relevant to the current study. On page 7 in Study Setting, the authors should note that this was a cluster-randomized trial and give the number of facilities involved.	Information has been added to the study setting about the number of facilities: “The trial was conducted in 24 intervention and 24 control facilities in Siaya County, Western Kenya.” We also added to the intervention description: “The Afya CCT intervention was a cluster randomised controlled trial, with equal allocation to intervention and control arms. The units of randomisation were level 2 or 3 health facilities (Dispensaries and Health Centres, respectively). The trial enrolled a total of 2522 women at intervention clinics and 2949 at control clinics.”
Table 1 and other parts of the manuscript allude to nurses receiving CCT, but this is not properly explained in the description of the intervention. Describe this more.	This has been added to the text about the intervention.
Table 3 should note that these numbers are by facility.	Facility has been added to Table 4 (formerly table 3).
The Results section spends ample space discussing nurses' views, but the Discussion fails to fully discuss these results. It is summarized that the intervention was acceptable to nurses, but the Results suggest that this was more nuanced, particularly in increased workload and the section "Ethicality." The findings in the latter section are resonant of those in another CCT among pregnant women in South Africa. See: Acceptability and feasibility of a financial incentive intervention to improve retention in	This is partly due to word count limitations, but we have expanded on these results relating to the nurses in the discussion. Thank you for the paper suggestion, we have included this as an example where similar findings were reported: “However other aspects of acceptability were lower from the perspective of nurses, including an extra burden of work created by the intervention, and ethical concerns related to cash transfers for maternal health visits. Some

HIV care among pregnant women in Johannesburg, South Africa. AIDS Care 2018 Apr;30(4):453-460	nurses viewed attendance at visits as part of the responsibility of the clients, and similar findings related to personal responsibility have been reported in a financial incentive intervention in Johannesburg, South Africa, to improve retention in HIV care among pregnant women [34]. “
Minor edits:  - Page 16, line 8: add a comma after "testing" - Page 19, line 57: "increase" not "increased" - Page 20, line 8: add "to" between "needed interact" 	These errors have been corrected.